# A Study on the Willingness of Industrial Ecological Transformation from China’s Zero Waste Cities Perspective

**DOI:** 10.3390/ijerph19159399

**Published:** 2022-07-31

**Authors:** Xing Li, Yongheng Fang, Fuzhou Luo

**Affiliations:** 1School of Management, Xi’an University of Architecture and Technology, Xi’an 710055, China; shinelixing@xauat.edu.cn (X.L.); 13557710701@163.com (F.L.); 2School of Public Administration, Xi’an University of Architecture and Technology, Xi’an 710055, China

**Keywords:** zero waste cities, industrial statistics, ecological transformation, questionnaire analysis

## Abstract

Based on the practice of a circular economy, China officially put forward the goal of building a “Zero Waste City” in 2018 and has formulated a series of measures to promote energy savings and emissions reduction in various sectors among which industry has received key attention due to its long-term high energy consumption and high pollution. The growth of an urban economy cannot be supported by industry, but the high energy consumption and high pollution of industry have become the keys to urban environmental management, and the need for ecological transformation of industry is very urgent. Based on the construction background of zero waste cities in China, this study analyzes the transformation factors of industrial producers’ willingness to make ecological transformation. The factors that influence industrial producers to make ecological transformation are perception of participation, perception of cost, perception of identity, and perception of government intervention. These factors have a positive moderating effect on the adoption of transformation strategies by industrial producers among which the perception of government involvement also plays a mediating role and has an important influence on the promotion of active ecological transformation by industrial producers.

## 1. Introduction

With the rapid development of global industrialization in the last century, which brought negative problems such as ecological destruction and resource depletion, human beings began to reflect on how to achieve a harmonious coexistence between human social development and natural ecology and thus put forward a sustainable development strategy and formed a set of theoretical systems [1]. The basic idea is to explore a way to meet the various production and living needs of modern social development without damaging the ecological environment and leaving negative impacts to future generations. On this basis, from the perspective of resource development and metabolism, on the one hand, there must be a limit to resource development, and it is necessary to increase the efficiency of resource use and the resource recycling rate and to find alternative energy sources [2]; on the other hand, when carrying out waste disposal, the damage to the natural ecological environment should be reduced as much as possible, and the discharged waste should be absorbed and degraded by nature without leaving problems for future generations; the most ideal state is “zero waste” so that resource production is always kept in the inner cycle.

Throughout the history of human development, it is obvious that the single pursuit of economic growth is contrary to the concept of sustainable development, especially the traditional high industrial growth stage, and the serious environmental damage and pollution problems arising from various types of industrial production still need to be remedied for a long period of time [3,4]. Therefore, in order to change the traditional single pursuit of economic growth as the goal, restore the damaged environment, reduce the impact on the environment, increase the resource cycle and alternative energy development, and other means to achieve the sustainable development goal of harmonious coexistence between humans and nature, countries around the world began to explore “zero waste” technology and practice methods [5]. In 2018, China officially proposed the concept of a “Zero Waste City,” which aims to lead the construction of a sustainable society by achieving efficient recycling and reducing solid waste emissions.

However, it is worth noting that from the perspective of producers, with the construction of zero waste cities, they are faced with the problem of ecological transformation of traditional production processes, which will certainly lead to higher costs and even lower production capacity for a certain period of time, affecting industrial producers to adopt active transformation strategies. In the construction of zero waste cities, local traditional industrial enterprises, as one of the pillar industries of a regional economy, will face many economic constraints and careful consideration of cost increase in the process of transformation, which is often the core element that affects their strategies on eco-transformation. Production supplier transformation initiative and participation will have important influences on urban construction waste and the government management level; management depth intervention factors will affect the production supplier’s performance and degree of ecological transformation cooperation. Through extensive research in this article, by using the survey data for analysis and calculation, the results show that government intervention, cost perception, identity perception, and participation intention have positive effects on the ecological transformation of traditional industrial enterprises in the background process of constructing a zero waste city.

This paper explores the factors and perceptions of Chinese industrial producers’ willingness to transform in the process of constructing zero waste cities, which can resolve the interest concerns of industrial producers and can provide ideas for the government to coordinate conflicts of interest and formulate management policies. The research follows the sequence of research hypothesis-research design-empirical analysis-conclusion and discussion, and it explores from the position of question raising-analysis of separation-problem solving. The research structure is shown in Figure 1.

## 2. Literature Review

A “Zero Waste City” is a leading initiative in Europe, which focuses on the reallocation of urban resources for recycling [6,7], such as water resources [8], mineral resources [9,10], and industrial solid waste disposal [11,12,13], according to the industrial structure and industrial characteristics of the city. In addition, many experts in management science have focused on the industrial development [14,15], industrial transformation [16,17], and industrial chain construction paths [18,19,20,21] of a “Zero Waste City”, and they have conducted extensive analysis and exploration of policies and measures. Especially in different countries, different social backgrounds as well as economic systems, and different regions have their own history of industrial development, therefore experts from various countries have proposed relevant urban management approaches for specific cities. By introducing social forces and adopting cooperation between government and enterprises to establish a benign industrial end-cycle, we aim to realize the material cycle through “industrial digestion” and realize the creation of a “Zero Waste” city [22,23,24,25,26].

China’s future urban resource cycle and metabolism need to rise to a new level, and the key issue is to break through the current bottleneck of low comprehensive resource utilization and focus on representative traditional industries to improve eco-efficiency and resource utilization. For example, the chemical industry, the iron and steel industry, the machinery manufacturing industry, the construction industry, etc., to improve the efficiency of material flow from the production chain [27,28,29]. It can be foreseen that future technical as well as non-technical issues of solving urban sustainable development problems are based on this goal, which require: exploring new technology applications and reducing energy consumption from a technical perspective; constructing relevant management models and realizing policy guarantees from a non-technical perspective and forming systematic and integrated solutions [30,31]; and strengthening regional and industrial linkages and synergies by landing in different cities, promoting relevant scientific and technological innovation and policy implementation, etc. [32,33].

At present, many studies on the relationship between traditional industry and urban economic development are mainly based on the perspective of industrial economics and environmental regulation, preferring economic management and public management. They mainly solve the problem of government governance [34,35,36], but do not start from the perspective of participants. Therefore, the dialectical relationship of conflict, dependence, and cooperation among different participants is analyzed from the endogenous perspective, and the subjective driving factors affecting the transformation of the traditional industrial ecology and the construction of a waste free city are not identified.

China’s transformation of its traditional industrial ecology started in the 1990s, showing explosive growth at the beginning of the 21st century. The achievements are very rich [37,38,39,40], especially for typical traditional industries, such as the automobile industry, the steel industry, and so on. Research of zero waste is based on the study of the circular economy of a city. The circular economy began in the United States in the 1960s, then the European Union, Australia, and other areas also undertook in-depth study, making the theoretical and the technological foundation of a zero waste city [41,42,43,44]. Now, China’s research on zero waste cities is a world leader, and research about relationships between zero waste cities and traditional industrial ecology transformation is not synchronized. Because China is in rapid economic growth, urban environmental pressure occurs in stages, so the ecological transformation of traditional industries is a driving force calling for more urgent research. Using the cross theories and research methods of economy and management, this paper points out that the transformation of traditional industrial ecology is a key problem in the construction of waste free cities in China, and it makes clear the relationship between urban management and industrial transformation.

After summarizing the literature, it was found that the current research on the relationship between the construction of zero waste cities and the ecological transformation of traditional industries is not deep enough, and it is mostly focused on the level of theory, planning, and practical analysis [45]. Since the concept of zero waste cities in China has not been proposed for a long time and the number of pilot cities is small, there is a lack of more in-depth research on the construction of zero waste cities and industrial ecological transformation, and there is no specific analysis of the factors affecting the willingness of industrial enterprises to transform, and the perception of industrial ecological transformation has not been comprehensively rationalized. Therefore, this paper focuses on the perception problem of industrial ecological transformation in the context of the construction of zero waste cities. Through the perspective of management science, the factors influencing industrial enterprises’ willingness to carry out ecological transformation are resolved, which complements the results of theoretical and empirical research on the correlation between the construction of zero waste cities and industrial transformation.

## 3. Research Design

### 3.1. Research Hypothesis

In order to study the factors influencing the adoption of ecological transformation strategies by industrial producers, five dimensions of perception variables were set. They are: industrial producers’ perception of participation; industrial producers’ perception of cost; industrial producers’ perception of identity; industrial producers’ perception of willingness to implement; and government intervention. The model is shown in Figure 2:

A total of six research hypotheses have been designed, named as H1–H6, and the specific hypotheses were as follows.

**Hypothesis** **1** **(H1).***The perceived participation of the industrial producer significantly and positively influences the willingness of the industrial producer to execute the transformation*.

**Hypothesis** **2** **(H2).***The cost perception of industrial production parties significantly and positively influences the willingness of industrial production parties to execute transformation*.

**Hypothesis** **3** **(H3).***The perceived identity of the industrial producer significantly and positively influences the willingness of the industrial producer to implement the transformation*.

**Hypothesis** **4** **(H4).***The perception of government involvement plays a positive moderating role, influencing industrial producers’ participation perception and their willingness to perform transformation*.

**Hypothesis** **5** **(H5).***The perception of government intervention plays a positive moderating role, influencing industrial producers’ cost perceptions and their willingness to execute transformation*.

**Hypothesis** **6** **(H6).***The perception of government intervention plays a positive moderating role, influencing industrial producers’ perceptions of identity and their willingness to perform transformation*.

### 3.2. Research Design

#### 3.2.1. Questionnaire Design and Variable Measurement

The questionnaire used in this study consists of two parts: the first part is the respondents’ basic personal information, which is investigated in four aspects: gender, age, monthly income, and the industry they work in; the second part of the questionnaire is the measurement scale questions of the five variables proposed in this study, which are measured on a 5-point Likert scale, ranging from “strongly disagree” to “strongly agree”, with corresponding values of “1” and “5”. The second part of the questionnaire presents the measurement scale questions of the five variables in this study, which are measured on a 5-point Likert scale, ranging from “strongly disagree” to “strongly agree”, with corresponding values of “1” and “5”. Table 1 shows the measurement scale questions for the five variables and the sources of the scales.

#### 3.2.2. Data Collection and Data Analysis Tools

The questionnaire survey was conducted from October 2021 to March 2022; 400 questionnaires were distributed; and 386 questionnaires were returned, with a total return rate of 95.75%, including 363 valid questionnaires and a valid return rate of 94.04%.

Because common method biases can have potentially serious effects on research findings, it is important to avoid this risk. The research relies on the designed questionnaire and forms self-administered data (data on independent and dependent variables emanate from the same respondent and the same measurement), therefore common method bias (CMB) may exist and affect the validity of survey results and the reliability of data [60]. According to the causes and solutions of CMB proposed by Podsakoff, P.M., this paper mainly tries to avoid common method bias from the following three perspectives [61,62]:Procedural Remedy

Cross-source data is used in the process of the questionnaire, that is the respondents include both managers who can make decisions on the behavior of enterprise eco-transformation and grass-roots employees who respond to the decision-making of enterprise eco-transformation so as to more comprehensively reflect the intention and perceptional factors of enterprise eco-transformation.

2.Technical Remedy

Firstly, the questionnaire was distributed in various forms, including email, telephone, face-to-face interview, and online questionnaire, to overcome non-response bias. Secondly, an anonymous approach was adopted to reduce the concerns of respondents and reduce method biases. Finally, the questionnaire explains that there is no correct or wrong answer to encourage answers to the questions that were as honest as possible.

3.Statistical Remedy

One of the most widely used research techniques to address the issue of common method variance is what has come to be called Harman’s one-factor (or single-factor) test. According to the basic information of the first part of the structured questionnaire, the sample population characteristics were filtered and a single factor test was conducted [63,64]; the results are shown in Table 2.

#### 3.2.3. Demographic Characteristics of Samples

Firstly, Table 2 shows the results of an analysis of the samples’ demographic characteristics:

As can be seen from Table 2, the majority of respondents in this survey are female, and in the age distribution, the two age groups of 30–35 and 36–40 account for the most respondents, 31.1% and 28.1%, respectively; in the monthly income distribution, more than half of the respondents have a monthly income of RMB 10,001–15,000, accounting for most of the respondents, and the proportion of both low-income group (below RMB 5000 yuan) and high-income group (above 20,000 yuan) is low. In the distribution of industries, 45.2% of the respondents were engaged in the construction industry, accounting for the highest percentage; 22% of the respondents were engaged in the automobile manufacturing industry, accounting for the second highest percentage, and only 0.6% of the respondents were engaged in other traditional industries. The skewness and kurtosis in Table 2 are tests of the formal research data, and it is usually considered that when the absolute value of skewness is less than 3 and the absolute value of kurtosis is less than 7, the sample meets the normal distribution. From Table 2, the absolute value of skewness is less than 3 and the absolute value of kurtosis is less than 7 for all measurement items, so it can be considered that the sample meets the normal distribution. In addition, the four basic information (gender, age, monthly income, and industry classification) and the original data of 20 questions A1–E4 in Table 1 have passed the variance test, which means that different basic samples show a consistency for all questions without difference, meeting the requirements of further data analysis.

In this study, two statistical analysis software, SPSS 21.0 (Version 21.0, IBM^®^, New York, NY, USA) and AMOS 10.0 (Version 10.0, IBM^®^, New York, NY, USA), were used for data coding and empirical analysis of the questionnaire data.

## 4. Case Study

### 4.1. Descriptive Statistical Analysis

The relationship model proposed in this study includes five variables, namely, willingness to perform, perception of participation, perception of cost, perception of identity, and perception of government involvement, with 20 question items. All the scales use a 5-point Likert scale with a maximum value of 5 and a minimum value of 1. All the variables are scored with higher scores corresponding to higher evaluation levels. The descriptive statistics of the 20 measurement items are shown in Table 3:

The large sample survey data of each measurement item in this study meets the threshold requirements, and further analysis can be conducted.

### 4.2. Questionnaire Reliability Analysis

The reliability test of a questionnaire is a test of the reliability and trustworthiness of the questionnaire, which is an indicator of the true extent of the data being tested, based on the consistency or stability of the results obtained from the test instrument. Reliability is divided into intrinsic reliability and extrinsic reliability. The intrinsic reliability test is to test whether a set of questions measure the same concept, and the consistency of the questions of the scale. In this study, Cronbach’s Alpha was used to measure the internal reliability of the questionnaire. The larger the coefficient in Cronbach’s Alpha measure, the higher the internal consistency of the questionnaire. The coefficients take values in the range of 0 to 1. The researcher concluded that a Cronbach’s Alpha coefficient greater than 0.9 indicates a high internal consistency of the questionnaire; a Cronbach’s Alpha coefficient between 0.8 and 0.9 indicates good internal consistency of the questionnaire; and a Cronbach’s Alpha coefficient between 0.7 and 0.8 indicates average internal consistency of the questionnaire. A Cronbach’s Alpha coefficient below 0.7 indicates that the internal consistency of the questionnaire is poor, and then the questionnaire is considered unsuitable as a research instrument. The internal consistency of the questionnaire was tested by examining the reliability of each part of the scale separately, and the test results are shown in the table. As can be seen from the table, Cronbach’s alpha of the scale is above 0.7, which indicates that the internal consistency of the questionnaire is relatively high, and the questionnaire can be used as a research tool for this study.

As shown in Table 4, the Cronbach’s Alpha for willingness to perform is 0.885; the Cronbach’s Alpha for perception of participation is 0.828; the Cronbach’s Alpha for perception of cost is 0.89; the Cronbach’s Alpha for perception of identity is 0.891; the Cronbach’s Alpha for perception of government involvement is 0.902; and the Cronbach’ s Alpha is 0.902. The Cronbach’s Alpha coefficients of each latent variable meet the basic criteria of greater than 0.7, and most of them are above 0.8. It can be seen that the questionnaire used in this study has good reliability. Besides, most of the CITCs between the observed variables and their latent variables are between 0.6 and 0.830, which meet the requirement of greater than 0.5. This indicates that the correlation coefficient CITC of each observed variable and the latent variable to which it belongs for each variable is more than 0.5, and most of them are between 0.6 and 0.830, which indicates that the latent variables of each question item are well set and the questionnaire has good reliability. At the same time, by excluding the observed variables, the specific practice is to delete each variable once. If there is no change in the reliability index after the deletion, the variable is considered to have good reliability for the measurement question.

### 4.3. Validity Analysis of Questionnaire

Questionnaire validity is broadly defined as the validity of a questionnaire scale, including content validity and construct validity. Content validity refers to the actuality and representativeness of the test or scale content and questions, while construct validity refers to the extent to which the scale can correctly measure theoretical constructs or traits. In the first step, exploratory factor analysis (EFA) is conducted to test the existence of common factors among the items, and the common factors extracted from the variables can better represent the structure of the scale; in the second step, confirmatory factor analysis (CFA) is conducted to test the fit between the measured variables and the factor constructs. In the third step, the convergent validity and discriminant validity of the variables were calculated to determine whether the scale had good construct validity.

#### 4.3.1. Exploratory Factor Analysis

Exploratory factor analysis is to measure the structural validity of the scale, which is to determine whether the measurement variables of each latent variable have stable consistency and structure, and it is the most commonly used index when evaluating the validity of the scale. In this paper, SPSS software is applied to test the composition of each dimension. When using factor analysis for validity analysis, the first step is to determine whether the conditions of factor analysis are satisfied, generally two conditions need to be satisfied, one is that the KMO value needs to be greater than 0.7; the second is that the significance of Bartlett’s test of sphericity is less than 0.05, and if these two conditions are satisfied, it means that there is a strong correlation between the observed variables, which is suitable for factor analysis, and the results are shown in Table 5.

The test results showed that the KMO test value for the survey data was 0.916, which is greater than 0.70, indicating that the questionnaire is suitable for factor analysis. The Bartlett’s sphericity test showed that the approximate chi-square value was 4636.623, which is a relatively large value with a significance probability of 0.000 (*p* < 0.01), so the null hypothesis of Bartlett’s sphericity test was rejected and the scale was considered as suitable for factor analysis and therefore has a better validity structure. As a statistical method, this result in Table 5 also proved that the data of this questionnaire passed the single-factor test (Harman’s one-factor test) and avoided common method bias (CMB).

The principal component analysis method was adopted to extract factors with eigenvalues greater than 1. As per the result shown in Table 6, a total of 5 common factors were extracted, and the cumulative sum of squared rotations was 74.303%, which was greater than 60%. After rotation by the orthogonal rotation method, the 20 question options can be categorized into 5 categories of factors, each item’s loadings are higher than 0.5, indicating that the extracted 5 factors contain more comprehensive information, and there is no doubt factor loadings are high; the observed variables are aggregated under each dimension according to the theoretical prediction. Combined with the calculated results in Table 6, it indicates that the scale selected in this paper has good construct validity.

#### 4.3.2. Confirmatory Factor Analysis

Validating factor analysis is a statistical analysis of survey data, and this method is used to test whether the relationship between a factor and the corresponding observed variable conforms to the theoretical relationship predetermined by the researcher. In this paper, we mainly use validated factor analysis to calculate the convergent validity and discriminant validity of each latent variable, and generally require the standardized loadings of the observed variables to be greater than 0.5 in order to consider the scale to have good construct validity; the combined reliability (*CR*) is also one of the criteria to judge the quality of the scale, reflecting whether all the observed variables in each latent variable consistently explains the latent variable. Generally, a combined reliability *CR* greater than 0.7 is required, indicating that all observed variables in each latent variable can consistently explain the latent variable. The formula is shown in the table below, where *λ* is the standardized factor loading of each observed variable of the value, and *θ* is the measurement error of the observed variable; the equation is as follows:(1)$$CR=\frac{(\sum \lambda {)}^{2}}{[(\sum \lambda {)}^{2}+\sum \left(\theta )\right]},\;\theta =1-\lambda^{2}$$

The mean variance extracted is to measure the convergent and discriminant validity of each latent variable, mainly to explain how much of the variance of the latent variable is from measurement error, and it is generally believed that the larger the mean variance extracted, the smaller the relative measurement error. According to the rule of thumb, if the mean variance extracted from the latent variable is greater than 0.5, and the arithmetic square root of the mean variance extracted of the latent variable is greater than the correlation coefficient between the latent variable and other latent variables, the scale has good convergent validity. The formula is shown in the table below, where *λ* is the standardized factor loading of each observed variable of the value, and *θ* is the measurement error of the observed variable; the equation is as follows:(2)$$AVE=\frac{\sum \lambda^{2}}{[\sum \lambda^{2}+\sum \left(\theta )\right]},\;\theta =1-\lambda^{2}$$

Validating factor analysis (CFA) was performed on the scales in this paper using AMOS 21.0. The validation factor model was established based on the results of exploratory factor analysis, and the suitability of the validation factor model constructed in this paper was judged by judging the structural equation fitting index. If the criteria are met, it means that the model constructed in this paper can effectively measure the relevant latent variables, as shown in Table 7:

According to the confirmatory factor model in Figure 3 and the mathematical analysis of confirmatory factors, the following index data can be obtained, as shown in Table 8:

In general, the ratio of cardinality to degrees of freedom should be greater than 1 and less than 3. A ratio greater than 3 indicates a poor fit and a ratio less than 1 indicates an overfit. The GFI (goodness of fit index) and AGFI (adjusted goodness of fit index) are the adjusted goodness of fit indices. NFI (Normed fit index) is the baseline fit index. If NFI is equal to 1 minus the model difference, the smaller the model difference. The closer the NFI value is to 1, the better the model fit. TLI (Tucker–Lewis index) is usually between 0 and 1, when TLI is equal to 1, it means the data fits the model completely. The common standard is 0.9 and the TLI of this paper is 0.965. RMSEA is the root mean squared of the asymptotic residuals, which is the ratio of the overall variance to the degrees of freedom, usually less than 0.08. The RMSEA of this paper is 0.048. In summary, all the indicators of the validation factor analysis have met the standard, and the overall fit of the model is good.

#### 4.3.3. Convergent Validity and Discriminant Validity Test

Convergence validity test

Convergent validity (CV) is defined as the classification obtained when two different measurement instruments are used to measure the same concept is highly correlated. In this study, convergent validity was examined by construct reliability (*CR*) and average variance extracted (*AVE*). Construct reliability is usually >0.7, and *AVE* > 0.5 is the criterion met.

The results of the validated factor analysis of the overall scale are shown in Table 9. The standardized factor loadings of the question items under the five variables of willingness to perform, perception of participation, perception of cost, perception of identity, and perception of government involvement are all above 0.5, indicating that each observed variable can explain its latent variables to a large extent. The combined reliability *CR* is greater than 0.8, which is significantly higher than the standard 0.7, so the observed variables under each dimension can explain the dimension well. The greater the *AVE* value, the greater the percentage of variance explained by the latent variable, and the smaller the relative measurement error, which is generally taken to be above 0.5. The above table shows that the *AVE* values are above the standard value of 0.5, which indicates that the scale has good convergent validity.

2.Discriminant validity

In order to ensure that there is a good differentiation between the different factors, the discriminant validity was calculated. In this study, the *AVE* was used to calculate the discriminant validity, and if the arithmetic square root of the *AVE* value for each factor was greater than the standardized correlation coefficient (i.e., the value below the diagonal in the table), then all factors in this study were judged to have discriminant validity, and the specific values are shown in Table 10.

From the above table, the arithmetic square root value of willingness to perform *AVE* is equal to 0.813 > 0.5; the arithmetic square root value of participation perception *AVE* is equal to 0.740 > 0.5; the arithmetic square root value of cost perception *AVE* is equal to 0.822 > 0.5; the arithmetic square root value of perception of identity *AVE* is equal to 0.820 > 0.5; and the arithmetic square root value of government intervention perception *AVE* is equal to 0.835 > 0.5. All of these values are greater than the lower diagonal values, indicating that the scale has good convergent and discriminant validity.

### 4.4. Model Test and Its Results

#### 4.4.1. Correlation Analysis

Correlation analysis can explain the degree of correlation between variables, but there are limitations in explaining causality. In order to further explain the causal relationship between variables, the article utilized Pearson correlation analysis to analyze the relationship between each variable in this study.

The data in Table 11 show that the correlation coefficients between willingness to implement and perceptions of participation, perceptions of cost, perceptions of identity, and perceptions of government involvement are 0.623, 0.556, 0.529, and 0.325, respectively, and the corresponding *p*-values are less than 0.01, which are statistically significant. It indicates that participation perception, cost perception, perception of identity, government intervention perception, and willingness to implement all have significant positive correlations.

#### 4.4.2. Structural Equation Model

Based on the theoretical model, a structural equation model was developed using AMOS 21.0, with participation perception, cost perception, and perception of identity as independent variables and willingness to perform as dependent variables, as shown in Figure 4.

The goodness-of-fit was calculated according to the set model and the calculated data are shown in Table 12:

In judging whether the structural equation model holds, it is mainly measured by the measurement of some fit indices. Among them, *χ*^2^/df is generally required to be less than 3, GFI is the fitness index, AGFI is the adjusted fitness index, NFI is the gauge fitness index, IFI is the value-added fitness index, CFI is the comparative fitness index, and these values are generally required to be greater than 0.9, which means that the model has a good fitness, but a value greater than 0.8 means that the model is acceptable. From the table below, *χ*^2^/df is 1.697 less than 3, GFI is 0.945 more than 0.8, AGFI is 0.924 more than 0.8, NFI is 0.954 more than 0.9, CFI is 0.981 more than 0.9, and RMSEA = 0.044 less than 0.08. It means the model fit is good and the model is acceptable. According to the criteria of the model fitting index, all the fitting indexes of the model meet the requirements, so the path of the model is analyzed. The result is shown in Table 13.

The standardized path coefficient of participation perception to willingness to execute was 0.455 (*t*-value = 5.728, *p* = 0.000 < 0.01), indicating that participation perception has a significant positive effect on willingness to execute. That is, the higher the perception of involvement, the higher the willingness to execute, so hypothesis H1 is accepted.

The standardized path coefficient of cost perception to willingness to execute is 0.222 (*t*-value = 3.453, *p* = 0.000 < 0.01), which indicates that cost perception has a significant positive effect on willingness to execute. That is, the higher the cost perception, the higher the willingness to execute, so hypothesis H2 is accepted.

The standardized path coefficient from perception of identity to willingness to execute is 0.192 (*t*-value = 3.203, *p* = 0.001 < 0.01), which indicates that perception of identity has a significant positive effect on willingness to execute. That is, the higher the perception of identity, the higher the willingness to execute, so hypothesis H3 is accepted.

#### 4.4.3. Adjustment Effect Test

The test of moderating utility was conducted using linear regression in SPSS 21.0, and four main models were constructed in which the dependent variable was willingness to perform. Model 1 introduces basic information: gender, age, monthly income, and industry engaged. Model 2 introduces the independent and moderating variables, and Model 3 introduces the interaction term.

The data in Table 14 show that: in model 1 control variables have no significant effect on willingness to perform; in model 2 the independent variable perception of involvement has a significant positive effect on willingness to perform (β = 0.624, t = 15.103); and in model 4 the regression coefficient of the interaction term between the independent and moderating variables is 0.149 (t = 3.541), which indicates that the interaction term has a significant effect on willingness to perform. The R2 of model 2 is 0.394 and the R2 of model 4 is 0.427, which is significantly higher, indicating the enhanced explanatory power of the model. The VIF value is less than 1.192, and there is no multicollinearity in the model. Therefore, it is proven that the moderating variable government involvement perception has a significant moderating effect on the impact of participation perception and willingness to perform, and the hypothesis is accepted.

The moderating effect of the moderating variable perceived government involvement in cost perception and willingness to perform was also verified by building a regression model.

The calculated results in Table 15 show that: in model 1 control variables have no significant effect on willingness to perform; in model 2 the independent variable cost perception has a significant positive effect on willingness to perform (β = 0.557, t = 12.649); and in model 4 the regression coefficient of the interaction term between the independent and moderating variables is 0.139 (t = 3.062), which indicates that the interaction term has a significant effect on willingness to perform. The R2 of model 2 is 0.314 and the R2 of model 4 is 0.356, which is significantly higher, indicating the enhanced explanatory power of the model. The VIF value is less than 1.239, and there is no multicollinearity in the model. Therefore, it is proven that the moderating variable perceived government involvement has a significant moderating effect on the impact of cost perception and willingness to perform, and the hypothesis is accepted.

The moderating effect of the moderating variable perceived government involvement in the perception of identity and willingness to perform was also verified by building a regression model.

The data in Table 16 show that: in model 1 control variables have no significant effect on willingness to perform; in model 2 the independent variable identity knowledge has a significant positive effect on willingness to perform (β = 0.544, t = 12.132); and in model 4 the regression coefficient of the interaction term between the independent and moderating variables is 0.153 (t = 3.413), which indicates that the interaction term has a significant effect on willingness to perform. The R2 of model 2 is 0.297 and the R2 of model 4 is 0.347, which is significantly higher, indicating the enhanced explanatory power of the model. The VIF value is less than 1.178, and the model does not have multicollinearity. Therefore, it is proven that the moderating variable perception of government involvement has a significant moderating effect on the influence of perception of identity and willingness to perform, and the hypothesis is accepted.

#### 4.4.4. Results of Research Hypothesis Testing

Based on the valid questionnaire data to empirically test the established research model, all hypotheses presented in the paper were verified. Results of data analysis showed that all hypotheses from H1 to H6 were accepted (hypotheses H1–H6 have shown in Section 3.1 before).

## 5. Conclusions

### 5.1. Results

Through extensive investigation combined with the questionnaire situation, the structural equation model was established to analyze the questionnaire, and finally it was found that the production supplier’s willingness to participate had a significantly positive impact on their willingness to execute the transformation; the production supplier’s cost perception had a significantly positive impact on their willingness to implement transformation; the production supplier’s perception of identity had a significant positive impact on their intention to execute the transition; perception of government intervention played a positive moderating role, affecting the supplier’s willingness to participate and their willingness to execute the transition; perception of government intervention played a positive moderating role, impacting the cost perception of suppliers and their intention to implement transformation; perception of government intervention played a positive moderating role in the effect of perception of production supplier’s identity on their intention to implement transformation. It shows that government intervention and mass support have positive regulating effects on the ecological transformation of traditional industry.

Traditional industry, especially heavy industry, has been an important support industry for the local economy for a long period of time, providing a huge boost to regional economies and improving regional infrastructure. However, the survey found that with the improvement of people’s living standards and the continuous enrichment of material and cultural life, urban residents are paying more and more attention to their residential environment, and there is a trend of more enthusiastic and active participation in it. In the past, traditional industries brought enormous pressure to the urban ecological environment due to high energy consumption and pollution, and there are significant negative externalities. When negative externalities lead to the inability to allocate resources effectively through the market, timely and scientific intervention by the government is particularly important. Especially for negative externalities and market failures, the government can take two kinds of measures: one is to utilize the administrative function for control, and the other is to utilize the economic role to participate in the form of incentives or subsidies.

The theoretical significance is to provide ideas for solving the cooperative mechanism of traditional industrial eco-transformation with a zero-waste mode. The construction of waste free cities is a long-term, complex, and dynamic process of constant adjustment. By exploring the problems existing in the eco-transformation of traditional industries under the background of construction of zero waste cities, this study can realize the goal of constructing low-carbon cities in China by putting forward the non-technical management methods in China’s urbanization process for various ecological problems (including ecological destruction, resource depletion, etc.) and ensuring that construction of zero waste cities and eco-transformation of traditional industries can be smoothly implemented and gradually promoted. Besides that, it also provides a referential model to achieve the goal of a zero-waste society in the future.

The practical significance of this paper is to identify the driving force for the transformation of the traditional industrial ecology in the process of building zero waste cities in China. The analysis reveals that government regulation, public recognition, and the cost perception of industrial enterprises will have a positive effect on the willingness to transform. Therefore, the multiparty effect should be fully utilized to promote industrial producers to actively carry out ecological transformation. Through the design of the traditional industrial eco-transformation intention questionnaire under the background of zero waste cities and statistical analysis, the factors to adopt the eco-transformation strategy for enterprises are clarified. The article has put forward a scientific and a reasonable solution, while promoting the building of zero waste cities at the same time; it will be an effective guarantee to achieve the goal of construction of zero waste cities in China, making the traditional industries’ eco-transformation proceed smoothly.

### 5.2. Research Limitation

Due to the insufficient coverage of the current research work and the short period of the construction of zero waste cities in China, a part of the public has an insufficient understanding of the construction of zero waste cities, and there are limitations in the number of samples and the depth of interviews. In addition, the design of the questionnaire survey is based on combing out from the literature research in consideration of which some hidden variables are not fully reflected. The research and conclusion are based on the current social cognition of the construction of zero waste cities and industrial eco-transformation in China; with the maturation and development of zero waste cities in the future, the research will continue to improve.

### 5.3. Future Research Direction

At present, various kinds of industries are researching ecological technology and are carrying out innovative exploration of ways to achieve waste reduction, zero waste, and low carbon technology. This paper analyzes the influencing factors and mechanisms of the transformation of the traditional industrial ecology under the background of zero waste city management, providing inspiration and reference for government, enterprises, and residents. However, the content is only for a single industry, not for the industrial chain to explore related management modes, and it is likely to cause a mismatch in the traditional industrial production chain upstream and downstream interface. Researching from the perspective of industry chain management in the future is necessary. In the future, the construction of a multiparty collaboration model and an institutional guarantee can be explored in depth so as to improve the initiative of industrial ecological transformation from the management path, and it will provide for the ecological transformation and sustainable development of relevant industries.

## Figures and Tables

**Figure 1 ijerph-19-09399-f001:**
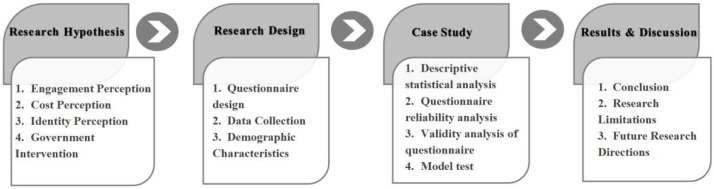
Research Path and Structure.

**Figure 2 ijerph-19-09399-f002:**
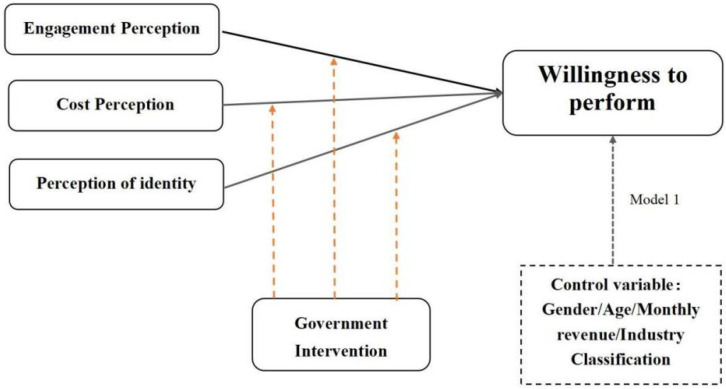
The perception model of industrial ecological transformation willingness for zero waste cities.

**Figure 3 ijerph-19-09399-f003:**
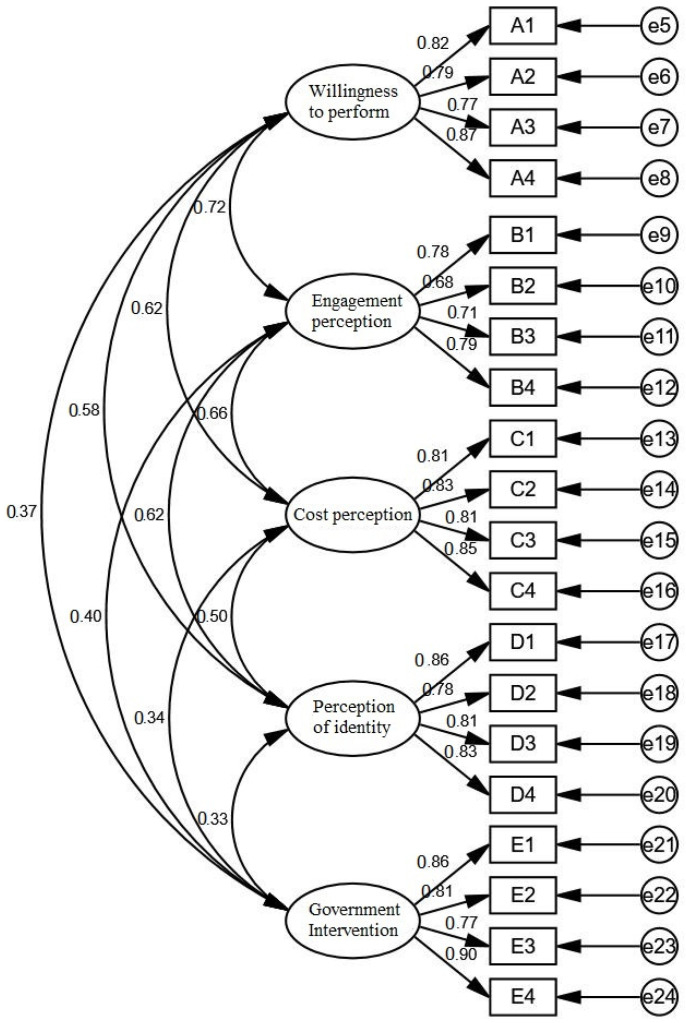
Validation factor model.

**Figure 4 ijerph-19-09399-f004:**
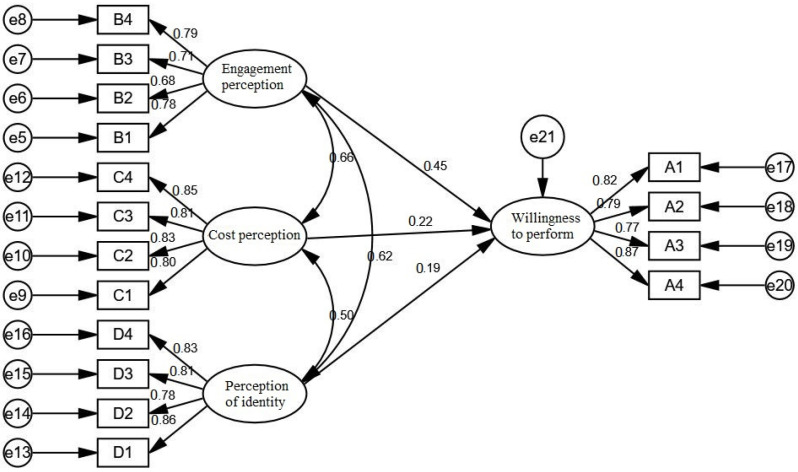
Structural equation model normalized path estimation.

**Table 1 ijerph-19-09399-t001:** Variable measurement scale.

Variable Name	No.	Measurement Question Items	Source
Willingness to perform	A1	The company will comply with the laws and regulations related to the construction of zero waste cities.	Weiyi H et al. (2019) [46], Pan L (2016) [47] Zhang et al. (2009) [48], Jiajuan Y et al. (2018) [49]
	A2	Enterprises are willing to assume the main responsibility for the construction of zero waste cities.	
	A3	Enterprises will actively carry out industrial technology upgrades to enhance product competitiveness.	
	A4	The company establishes an environmental management system that meets the requirements of zero waste cities.	
Engagement Perception	B1	Enterprises take the initiative to care about the industry and “zero waste cities” construction relevance.	Haitao W (2018) [50], Nan X et al. (2020) [51].
	B2	Enterprises are willing to actively participate in the ecological transformation of traditional industries and the construction of “zero waste city”.	
	B3	I can make constructive suggestions for the construction of zero waste cities.	
	B4	Corporate initiatives to fulfill social and environmental responsibility.	
Cost perception	C1	Companies will actively introduce green human capital.	Aditi Sengupta (2015) [52] Guanghua S et al. (2019) [53]
	C2	Enterprises are willing to bear the cost of pollution control.	
	C3	Companies can take the risk of ecological transformation.	
	C4	Companies are willing to invest costs in eco-transformation R & D and innovation.	
Perception of identity	D1	The ecological transformation of enterprises can gain the support of local residents.	Pengjie X (2018) [54], Rosa Maria Dangelico et al. (2017) [55], Xing L et al. (2022) [56].
	D2	Traditional industrial “Green floating behavior” ^1^ will affect market reputation.	
	D3	The ecological transformation of enterprises can gain the recognition of consumers.	
	D4	The ecological transformation of enterprises is an important part of the construction of zero waste cities.	
Government Intervention Perception	E1	The government is concerned about the ecological transformation of traditional industries in relation to the construction of zero waste cities.	Qi G et al. (2011) [57] Alidrisi H (2021) [58], Menglin Xing et al. (2018) [59].
	E2	I think government intervention has promoted energy saving and emission reduction in traditional industries.	
	E3	Government involvement can facilitate traditional industries to invest in green innovation.	
	E4	I think the government’s involvement has accelerated the process of building zero waste cities.	

^1^ Green floating behavior: companies promote their environmental behavior through external public opinion but in fact do not really take environmental measures.

**Table 2 ijerph-19-09399-t002:** Demographic characteristics of the sample.

Index	Options	Frequency	Standard Deviation	Skewness	Kurtosis
Gender	Male	150	0.493	−0.352	1.124
	Female	213			
Age	Under 30 years old	55	1.048	0.002	2.003
	30–35 years old	113			
	36–40 years old	102			
	41–46 years old	88			
	Over 46 years old	5			
Monthly income	Less than 5000 RMB	8	0.790	0.134	3.312
	5000–10,000 RMB	77			
	10,001–15,000 RMB	197			
	15,001–20,000 RMB	69			
	Above 20,000 RMB	12			
Industry Classification	Steel Industry	48	0.964	0.330	2.215
	Construction	164			
	Automotive Manufacturing	80			
	Textile Industry	69			
	Food industry *	1			
	Rubber industry *	1			

* The food industry and rubber industry were combined into other traditional industries in the questionnaire.

**Table 3 ijerph-19-09399-t003:** Descriptive statistical analysis of the sample.

Variables	No.	N	Minimal Value	Maximum Value	Average Value	Standard Deviation	Skewness	Kurtosis
Willingness to perform	A1	363	1	5	3.87	1.03	−0.60	−0.33
	A2	363	1	5	3.83	1.01	−0.49	−0.40
	A3	363	1	5	3.82	1.00	−0.53	−0.28
	A4	363	1	5	3.88	1.04	−0.60	−0.38
Engagement Perception	B1	363	1	5	3.94	1.16	−0.84	−0.32
	B2	363	1	5	3.89	1.04	−0.76	0.04
	B3	363	1	5	3.80	1.05	−0.66	−0.36
	B4	363	1	5	3.98	1.11	−0.77	−0.44
Cost perception	C1	363	1	5	3.59	1.21	−0.42	−0.83
	C2	363	1	5	3.60	1.26	−0.54	−0.80
	C3	363	1	5	3.53	1.20	−0.42	−0.80
	C4	363	1	5	3.62	1.24	−0.36	−1.05
Perception of identity	D1	363	1	5	3.46	1.09	−0.17	−0.75
	D2	363	1	5	3.35	1.07	−0.02	−0.83
	D3	363	1	5	3.50	1.13	−0.08	−1.01
	D4	363	1	5	3.37	1.03	−0.14	−0.43
Government Intervention Perception	E1	363	1	5	3.85	0.94	−0.67	−0.06
	E2	363	1	5	3.72	0.95	−0.49	−0.08
	E3	363	1	5	3.74	0.93	−0.47	−0.05
	E4	363	1	5	3.80	0.93	−0.68	0.33

**Table 4 ijerph-19-09399-t004:** Reliability analysis of variables in the questionnaire.

Variables	Title Item	CITC	Excluded Observed Cronbach’s Alpha Values	Cronbach’s Alpha Value
Willingness to perform	A1	0.742	0.855	0.885
	A2	0.735	0.857	
	A3	0.716	0.864	
	A4	0.803	0.831	
Engagement Perception	B1	0.682	0.770	0.828
	B2	0.610	0.802	
	B3	0.636	0.791	
	B4	0.690	0.766	
Cost perception	C1	0.749	0.867	0.893
	C2	0.767	0.861	
	C3	0.745	0.869	
	C4	0.794	0.851	
Perception of identity	D1	0.790	0.849	0.891
	D2	0.731	0.871	
	D3	0.757	0.862	
	D4	0.767	0.858	
Government Intervention Perception	E1	0.785	0.871	0.902
	E2	0.763	0.879	
	E3	0.741	0.886	
	E4	0.830	0.854	

**Table 5 ijerph-19-09399-t005:** Inspection of KMO and Bartlett.

Test		Result
Kaiser–Meyer–Olkin Measurement of Sampling Adequacy.		0.916
Bartlett’s Test of Sphericity	Approximate chi-square	4636.623
	df	190
	Sig.	0.000

**Table 6 ijerph-19-09399-t006:** Factor rotation matrix.

Variables	Title Item	Ingredients				
		1	2	3	4	5
Willingness to perform	A1				0.751	
	A2				0.756	
	A3				0.737	
	A4				0.816	
Engagement Perception	B1					0.722
	B2					0.701
	B3					0.712
	B4					0.709
Cost Perception	C1		0.797			
	C2		0.793			
	C3		0.768			
	C4		0.833			
Perception of identity	D1			0.812		
	D2			0.803		
	D3			0.820		
	D4			0.791		
Government Intervention Perception	E1	0.844				
	E2	0.837				
	E3	0.847				
	E4	0.886				
Eigenvalue		8.365	2.398	1.731	1.315	1.051
Variance contribution rate		15.775	15.528	15.336	14.783	12.881
Cumulative contribution rate		74.303%				

**Table 7 ijerph-19-09399-t007:** Main evaluation indexes and standards of overall suitability of the model.

Statistical Inspection Quantity	Adaptation Criteria or Thresholds
Cardinality of freedom ratio (NC value)	1 < NC < 3, Adaptation good;
GFI	>0.8
AGFI	>0.8
IFI	>0.9
CFI	>0.9
TLI	>0.9
IFI	>0.9
NFI	>0.9 Adaptation good,
RMSEA	<0.05 Adaptation good, <0.08 Adaptive rational

**Table 8 ijerph-19-09399-t008:** Model fitting metrics.

Index	X^2^/df	GFI	AGFI	NFI	IFI	TLI	CFI	RMSEA
Statistical values	1.847	0.926	0.902	0.938	0.970	0.965	0.970	0.048
Reference value	<3	>0.8	>0.8	>0.9	>0.9	>0.9	>0.9	<0.08
Achievement of standards	Achieve	Achieve	Achieve	Achieve	Achieve	Achieve	Achieve	Achieve

**Table 9 ijerph-19-09399-t009:** Results of convergent validity analysis.

Variables	Title Item	Standardized Factor Loadings	Standard Error	t-Value	*p*	*CR*	*AVE*
Willingness to perform	A1	0.821				0.886	0.661
	A2	0.791	0.056	16.872	***		
	A3	0.768	0.056	16.214	***		
	A4	0.867	0.056	19.015	***		
Engagement Perception	B1	0.777				0.829	0.549
	B2	0.682	0.061	12.741	***		
	B3	0.710	0.062	13.310	***		
	B4	0.788	0.065	14.861	***		
Cost perception	C1	0.806				0.893	0.677
	C2	0.828	0.061	17.458	***		
	C3	0.808	0.059	16.921	***		
	C4	0.848	0.060	17.982	***		
Perception of identity	D1	0.859				0.892	0.674
	D2	0.781	0.051	17.403	***		
	D3	0.808	0.053	18.311	***		
	D4	0.833	0.048	19.148	***		
Government Intervention Perception	E1	0.858				0.902	0.698
	E2	0.805	0.051	18.591	***		
	E3	0.770	0.051	17.365	***		
	E4	0.902	0.048	21.951	***		

Note: *** represents < 0.001.

**Table 10 ijerph-19-09399-t010:** Discriminant validity analysis test.

	Willingness to Perform	Engagement Perception	Cost Perception	Perception of Identity	Government Intervention Perception
Willingness to perform	**0.813**				
Engagement Perception	0.721	**0.740**			
Cost perception	0.619	0.660	**0.822**		
Perception of identity	0.585	0.618	0.499	**0.820**	
Government Intervention Perception	0.368	0.404	0.341	0.333	**0.835**

Note: The bold font is the arithmetic square root of *AVE*.

**Table 11 ijerph-19-09399-t011:** Correlation analysis results.

	Willingness to Perform	Engagement Perception	Cost Perception	Perception of Identity	Government Intervention Perception
Willingness to perform	1				
Engagement Perception	0.623 **	1			
Cost perception	0.556 **	0.575 **	1		
Perception of identity	0.529 **	0.527 **	0.440 **	1	
Government Intervention Perception	0.325 **	0.351 **	0.314 **	0.303 **	1
Average value	3.85	3.90	3.58	3.42	3.78
Standard deviation	0.88	0.89	1.07	0.94	0.82

** Significantly correlated at the 0.01 level (bilaterally).

**Table 12 ijerph-19-09399-t012:** Model fit goodness of fit index.

Reference Indicators	Evaluation Criteria	Statistical Values	Model Adaptation Judgment
$\chi^{2}$/df	Between 1 and 3 is ideal	1.697	Yes
AGFI	Greater than 0.8, the closer to 1 the higher the suitability	0.924	Yes
GFI	Greater than 0.8, the closer to 1 the higher the suitability	0.945	Yes
TLI	Greater than 0.9, the closer to 1 the higher the fitness	0.976	Yes
NFI	Greater than 0.9, the closer to 1 the higher the fitness	0.954	Yes
CFI	Greater than 0.9, the closer to 1 the higher the fitness	0.981	Yes
RMSEA	Less than 0.08	0.044	Yes

**Table 13 ijerph-19-09399-t013:** Path coefficients between the variables.

Paths	Non-Normalized Path Coefficient	Standard Error S.E.	C.R.	*p*	Standardized Path Coefficient	Assumptions
Engagement Perception positively affects Willingness to perform	0.426	0.074	5.728	***	0.455	Accept
Cost perception positively affects Willingness to perform	0.193	0.056	3.453	***	0.222	Accept
Identity Perception positively affects Willingness to perform	0.174	0.054	3.203	0.001	0.192	Accept

Note: *** represents < 0.001.

**Table 14 ijerph-19-09399-t014:** Moderating role of perceived government involvement in perception of participation and willingness to perform.

Models		Model 1	Model 2	Model 3	Model 4
		β	β	β	β
Control variables	Gender	0.040	0.064	0.064	0.072
	Age	−0.058	−0.034	−0.036	−0.020
	Monthly income	−0.029	−0.027	−0.022	−0.025
	Industry	0.040	0.011	0.013	0.008
Independent variable	Engagement Perception		0.624 ***	0.581 ***	0.599 ***
Adjustment variables	Government Intervention Perception			0.122 **	0.153 **
Interaction items	Participation in Perception & Government Intervention in Perception				0.149 ***
R^2^		0.007	0.394	0.407	0.427
Adjusted R^2^		−0.004	0.386	0.397	0.416
F		0.614	46.424 ***	40.715 ***	37.820 ***
VIF value		≤1.007	≤1.008	≤1.147	≤1.192

Note: N = 363, ** represents < 0.01, *** represents < 0.001.

**Table 15 ijerph-19-09399-t015:** Moderating role of perceived government involvement in cost perception and willingness to perform.

Models		Model 1	Model 2	Model 3	Model 4
		β	β	β	β
Control variables	Gender	0.040	0.031	0.035	0.038
	Age	−0.058	−0.015	−0.018	−0.015
	Monthly income	−0.029	−0.009	−0.004	0.002
	Industry	0.040	0.060	0.059	0.056
Independent variable	Cost perception		0.557 ***	0.504 ***	0.508 ***
Adjustment variables	Government Intervention Perception			0.168 ***	0.214 ***
Interaction items	Cost Perception &Government Intervention Perception				0.139 **
R^2^		0.007	0.314	0.339	0.356
Adjusted R^2^		−0.004	0.305	0.328	0.344
F		0.614	32.706 ***	30.490 ***	28.089 ***
VIF value		≤1.007	≤1.013	≤1.119	≤1.239

Note: N = 363, ** represents < 0.01, *** represents < 0.001.

**Table 16 ijerph-19-09399-t016:** Moderating role of perceived government involvement in perception of identity and willingness to perform.

Models		Model 1	Model 2	Model 3	Model 4
		β	β	β	β
Control variables	Gender	0.040	0.098	0.095 *	0.112
	Age	−0.058	−0.060	−0.059	−0.050
	Monthly income	−0.029	0.020	0.023	0.025
	Industry	0.040	0.066	0.065	0.064
Independent variable	Perception of identity		0.544 ***	0.490 ***	0.493 ***
Adjustment variables	Government Intervention Perception			0.180 ***	0.220 ***
Interaction items	Perception of recognition & perception of government involvement				0.153 **
R^2^		0.007	0.297	0.326	0.347
Adjusted R^2^		−0.004	0.287	0.315	0.335
F		0.614	30.127 ***	28.701 ***	27.001 ***
VIF value		≤1.007	≤1.022	≤1.124	≤1.178

Note: N = 363, * represents < 0.05, ** represents < 0.01, *** represents < 0.001.

## Data Availability

Not applicable.

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
