# Peer review of "A Study on the Willingness of Industrial Ecological Transformation from China’s Zero Waste Cities Perspective"

_ijerph, 2022, doi:10.3390/ijerph19159399_

Round 1

Reviewer 1 Report

Dear authors, congratulations for the work presented.

Below are my considerations and suggestions for improvement.

In the final part of the introduction, the authors should highlight, in a very summary way, the conclusions reached in this study.

The introduction must end with a paragraph that informs the structure that the work follows from this moment on. These two aspects will be the most relevant study for the reader.

In the literature review, there are no contradictory or opposing theories to the line of thought that the authors defend in their study. This leads to a more difficult perception of the problem under study. I suggest that the authors invest more in this point, in order to make their study more relevant.

The methodology followed is sufficiently detailed and suitable for the type of study.

The results are correctly interpreted and the conclusions are in line with the results.

Conclusions are scarce. Authors should put the contributions of this article to theory and practice, as this will make this article more relevant to the reader.

The limitations of the study and clues for future investigations should also be added.

Good luck.

Author Response

RE:Manuscript ID: ijerph-1773512 (Title: Study on the Willingness of Industrial Ecological Transformation from China's Zero Waste Cities perspective)

We would like to thank for giving us the opportunity to revise our manuscript.

We thank the editor and reviewers for their careful read and thoughtful comments on previous draft. We have carefully taken their comments into consideration in preparing our revision, which has resulted in a paper that is clearer, more compelling, and broader. The following summarizes how we responded to the comments of editor and reviewers .

Below is our response to their comments.

Thank you for your consideration and look forward to your reply soon.  Please contact me regarding to this manuscript.

Sincerely yours,

Xing Li on behalf of the authors.

e-mail:shinelixing@xauat.edu.cn

Revision_authors response:

Reviewers 1

Dear authors, congratulations for the work presented.

Below are my considerations and suggestions for improvement.

In the final part of the introduction, the authors should highlight, in a very summary way, the conclusions reached in this study.

The introduction must end with a paragraph that informs the structure that the work follows from this moment on. These two aspects will be the most relevant study for the reader.

RE: We have added a summary of the conclusion, the path and structure of the paper research in the end of introduction.

In the literature review, there are no contradictory or opposing theories to the line of thought that the authors defend in their study. This leads to a more difficult perception of the problem under study. I suggest that the authors invest more in this point, in order to make their study more relevant.

RE: In the final part of literature review, we have added discussion section about the differences of results in previous studies, and have pointed out the connections and differences between this study and others’ research to make the necessity and innovation of this study more concise.

The methodology followed is sufficiently detailed and suitable for the type of study.The results are correctly interpreted and the conclusions are in line with the results.

Conclusions are scarce. Authors should put the contributions of this article to theory and practice, as this will make this article more relevant to the reader.

The limitations of the study and clues for future investigations should also be added.

RE: We have added theoretical and managerial implications in 5.1 part. We also have added 5.2 part to illustrate the limitations of this study, which improving the rigor of the research. In addition, 5.3 part have been added as the future research direction, and the future research prospect of urban management and industrial transformation is proposed.

Reviewer 2 Report

The article is interesting, the research design is solid, and the researched problem has scientific potential. However, some problems need to be solved:

 1. Literature review should include more recent sources (2018-2021) and be enriched with relevant references. 

2. The use of self-administered questionnaires can generate a problem that may affect the relevance of the research: the bias effect or common method bias - CMB (see: Podsakoff PM, MacKenzie SB, Lee, JY, Podsakoff NP. Common method biases in behavioral research: A critical review of the literature and recommended remedies. Journal of Applied Psychology. 2003; 88(5):879-903.). Such problems arise when data on independent and dependent variables emanate from the same respondent and the same measurement scale exists throughout the questionnaire. Authors must take action to prevent common method bias - CMB (e.g., https://doi.org/10.3390/ijerph182312387) 

3. There is a need for a discussion section built in the context of dialogue with researchers in the literature review. 

4. In my opinion, a section of conclusions that includes theoretical and managerial implications, research limitations, and future research directions would be helpful. The current form of the conclusions briefly repeats what was stated in the results and discussion sections.

The article presents some scientific value and can be published after carefully reviewing the reported issues.

Author Response

RE:Manuscript ID: ijerph-1773512 (Title: Study on the Willingness of Industrial Ecological Transformation from China's Zero Waste Cities perspective)

We would like to thank for giving us the opportunity to revise our manuscript.

We thank the editor and reviewers for their careful read and thoughtful comments on previous draft. We have carefully taken their comments into consideration in preparing our revision, which has resulted in a paper that is clearer, more compelling, and broader. The following summarizes how we responded to the comments of editor and reviewers .

Below is our response to their comments.

Thank you for your consideration and look forward to your reply soon.  Please contact me regarding to this manuscript.

Sincerely yours,

Xing Li on behalf of the authors.

e-mail:shinelixing@xauat.edu.cn

Revision_authors response:

Reviewers 2:

The article is interesting, the research design is solid, and the researched problem has scientific potential. However, some problems need to be solved:

  1. Literature review should include more recent sources (2018-2021) and be enriched with relevant references. 

RE: We have added more recent source (2018-2022) to enrich the relevant references, and some references with weak correlation have been deleted.

  1. The use of self-administered questionnaires can generate a problem that may affect the relevance of the research: the bias effect or common method bias - CMB (see: Podsakoff PM, MacKenzie SB, Lee, JY, Podsakoff NP. Common method biases in behavioral research: A critical review of the literature and recommended remedies. Journal of Applied Psychology. 2003; 88(5):879-903.). Such problems arise when data on independent and dependent variables emanate from the same respondent and the same measurement scale exists throughout the questionnaire. Authors must take action to prevent common method bias - CMB (e.g., https://doi.org/10.3390/ijerph182312387) 

RE: We have carefully read the recommended literature according to the reviewer's suggestion and consulted other relevant literature. This paper mainly avoids Common Method Bias(CMB) from the following three perspectives: 

  1. Procedural Remedy

Cross-source data is used in the process of the questionnaire, that is the respondents include both managers who can make decisions on the behavior of enterprise eco-transformation and grass-roots employees who respond to the decision-making of enterprise eco-transformation, so as to more comprehensively reflect the intention and perceptional factors of enterprise eco-transformation.

  1. Technical Remedy

Firstly, the questionnaire was distributed in various forms, including email, telephone, face-to-face interview and online questionnaire, to overcome non-response bias. Secondly, anonymous approach is adopted to reduce the concerns of respondents and reduce method biases; Finally, the questionnaire explains that there is no correct or wrong answer to make their answer the questions as honestly as possible.

  1. Statistical Remedy

One of the most widely used techniques that has been used by researchers to address the issue of common method variance is what has come to be called Harman's one-factor (or single-factor) test. According to the basic information of the first part of the structured questionnaire, the sample population characteristics were sorted out and single factor test was conducted, the results are shown in Table 2 of manuscript.

The specific content above have been added in section 3.2.1 of this paper, and we have added a description of the results of statistical methods to avoid CMB after Table 5 in Section 4.3.1.

  1. There is a need for a discussion section built in the context of dialogue with researchers in the literature review. 

RE: In the final part of literature review, we have added discussion part about the differences of results in previous studies, and have pointed out the connections and differences between this study and others’ research to make the necessity and innovation of this study more concise.

  1. In my opinion, a section of conclusions that includes theoretical and managerial implications, research limitations, and future research directions would be helpful. The current form of the conclusions briefly repeats what was stated in the results and discussion sections.

RE: We have added theoretical and managerial implications in 5.1 part. We also have added 5.2 part to illustrate the limitations of this study, which improving the rigor of the research. In addition, 5.3 part have been added as the future research direction to propose the future research prospect of urban management and industrial transformation.

The article presents some scientific value and can be published after carefully reviewing the reported issues.

Reviewer 3 Report

The authors of this study investigated the factors and perceptions that influence Chinese industrial producers' willingness to transform in the construction of zero-waste cities, intending to sort out industrial producers' conflict of interest concerns and provide ideas for the government to coordinate conflict of interest and formulate management policies. The following are some minor points to consider:

1. The author should carefully check the manuscript such as typos and references. Some error in the Reference source was found in this version.

2. The author should clarify the other traditional industries for the questionnaire survey.

3. Why does the questionnaire survey mainly focus on the construction? 

Author Response

RE:Manuscript ID: ijerph-1773512 (Title: Study on the Willingness of Industrial Ecological Transformation from China's Zero Waste Cities perspective)

We would like to thank for giving us the opportunity to revise our manuscript.

We thank the editor and reviewers for their careful read and thoughtful comments on previous draft. We have carefully taken their comments into consideration in preparing our revision, which has resulted in a paper that is clearer, more compelling, and broader. The following summarizes how we responded to the comments of editor and reviewers .

Below is our response to their comments.

Thank you for your consideration and look forward to your reply soon.  Please contact me regarding to this manuscript.

Sincerely yours,

Xing Li on behalf of the authors.

e-mail:shinelixing@xauat.edu.cn

Revision_authors response:

Reviewers 3:

The authors of this study investigated the factors and perceptions that influence Chinese industrial producers' willingness to transform in the construction of zero-waste cities, intending to sort out industrial producers' conflict of interest concerns and provide ideas for the government to coordinate conflict of interest and formulate management policies. The following are some minor points to consider:

  • The author should carefully check the manuscript such as typos and references. Some error in the Reference source was found in this version.

RE: The authors thank the reviewers for their carefulness. We have proofread the whole paper for many times, corrected the errors which have been found, adjusted some references.We also have deleted some references with weak correlation, and added more recent source (2018-2022) to enhanced the literature support of this paper.

  • The author should clarify the other traditional industries for the questionnaire survey.

RE: Other traditional industries in the questionnaire survey are rubber industry and food industry, which are clarified in Table 2 of the article.

  • Why does the questionnaire survey mainly focus on the construction? 

RE: China's zero waste city plan was put forward in 2018, it is a very short time from then until now,so it is still in its infancy., We found that many traditional industrial enterprises and the public do not have a deep understanding of the zero waste city in the preliminary interview process. Therefore, we mainly focus on construction of zero waste cities in questionnaire design. This not only reflects the current implementation of China's zero waste  city plan objectively and reasonably , but also provides a basis for other researches on the operation process of zero waste city in the future.

Reviewer 4 Report

The authors of the article raised a very important issue regarding zero waste cities in China. I think that the article brings new information to the issues related to the solutions for the reduction of energy consumption and rational use of the Earth's resources, in the era of rapid climate change. The only drawback of the work is the small number of references to literature.

Author Response

RE:Manuscript ID: ijerph-1773512 (Title: Study on the Willingness of Industrial Ecological Transformation from China's Zero Waste Cities perspective)

We would like to thank for giving us the opportunity to revise our manuscript.

We thank the editor and reviewers for their careful read and thoughtful comments on previous draft. We have carefully taken their comments into consideration in preparing our revision, which has resulted in a paper that is clearer, more compelling, and broader. The following summarizes how we responded to the comments of editor and reviewers .

Below is our response to their comments.

Thank you for your consideration and look forward to your reply soon.  Please contact me regarding to this manuscript.

Sincerely yours,

Xing Li on behalf of the authors.

e-mail:shinelixing@xauat.edu.cn

Revision_authors response:

Reviewers 4:

The authors of the article raised a very important issue regarding zero waste cities in China. I think that the article brings new information to the issues related to the solutions for the reduction of energy consumption and rational use of the Earth's resources, in the era of rapid climate change. The only drawback of the work is the small number of references to literature.

RE: The authors thank the reviewers for their carefulness. We have deleted some references with weak correlation, and added more recent source (2018-2022) to enhanced the literature support of this paper.

Round 2

Reviewer 1 Report

Congratulations for the work presented.

Good luck.

Reviewer 2 Report

The paper can be published in current version.